

# 1  Earthquake-induced landslides in Haiti: analysis of

# 2  seismotectonic and possible climatic influences

Hans-Balder Havenith[1], Kelly Guerrier[2], Romy Schlögel[1,3], Anika Braun[4], Sophia
Ulysse[2,5], Anne-Sophie Mreyen[6], Karl-Henry Victor[2], Newdeskarl Saint-Fleur[2], Lena
Cauchie[1], Dominique Boisson[2], Claude Prépetit[5]
[1]University of Liege, Department of Geology, Georisk and Environment, Liege, 4000, Belgium
[2]Université d'Etat d'Haïti, Faculté des Sciences, LMI-CARIBACT, URGéo, Port-au-Prince, Haiti
[3]Centre Spatial de Liège, Liege, 4000, Belgium
[4]TU Berlin, Faculty VI Planning Building, Environment Department of Engineering Geology, Berlin,
1587, Germany
[5]Unité Technique de Sismologie, Bureau des Mines et de l'Energie, Port-au-Prince, Delmas 31, Haiti
[6]University of Liege, Department of Urban & Environmental Engineering, Applied Geophysics, Liege,
4000, Belgium
Correspondence to: Hans-Balder Havenith (hb.havenith@uliege.be)





**Abstract.** First analyses of landslide distribution and triggering factors are presented for the region
affected by the August, 14, 2021, earthquake (Mw=7.2) in the Nippes Department, Haiti. Landslide
mapping was mainly carried out by comparing pre- and post-event remote imagery (~0.5 -1-m resolution)
available on Google Earth Pro® and Sentinel-2 (10-m resolution) satellite images. The first covered
about 50% of the affected region (for post-event imagery and before completion of the map in January
2022), the latter were selected to cover the entire potentially affected zone. On the basis of the completed
landslide inventory, comparisons are made with catalogues compiled by others both for the August 2021
and the January 2010 seismic events, including one open inventory (by the United States Geological
Survey) that was also used for further statistical analyses. Additionally, we studied the pre-2021
earthquake slope stability conditions. These comparisons show that the total number of landslides
mapped for the 2021 earthquake (7091) is larger than the one recently published by another research
team for the same event, but it is also clearly smaller than the one observed by two other research teams
for the 2010 earthquake (e.g., 23,567, for the open inventory). However, these apparently fewer
landslides triggered in 2021 cover much wider areas of slopes (>80 km$^2$) than those induced by the 2010
event (~25 km$^2$ – considering the open inventory). A simple statistical analysis indicates that the lower
number of 2021-landslides can be explained by the missing detection of the smallest landslides triggered
in 2021, partly due to the lower resolution imagery available for most of the areas affected by the recent
earthquake; this is also confirmed by an inventory completeness analysis based on size-frequency
statistics. The much larger total area of landslides triggered in 2021, compared to the 2010 earthquake,
can be related to different physical reasons: a) the larger earthquake magnitude in 2021; b) the more
central location of the fault segment that ruptured in 2021 with respect to coastal zones; c) and possible
climatic preconditioning of slope instability in the 2021-affected area. These observations are supported
by (1) a new pre-2021 earthquake landslide map, (2) rainfall distribution maps presented for different
periods (including October 2016 - when Hurricane Matthew had crossed the western part of Haiti),
covering both the 2010 and 2021 affected zones, as well as (3) shaking intensity prediction maps.
**1   Introduction**
This paper presents a first overview of landslides induced by the August 14, 2021, Nippes (Haiti)





earthquake. The epicenter (18.434° N / 73.482° W, hypocentral depth of 10 km) of this event is located
in the western part of the southern Haitian peninsula (see Unites States Geological Survey, USGS,
Earthquake Hazard Program page, earthquake.usgs.gov, presenting first information on the 2021 M 7.2
- Nippes, Haiti, event). Similar to the January 12, 2010, earthquake, the epicenter is located near the
surface expression of the Enriquillo-Plantain-Garden Fault (EPGF) that crosses the peninsula from east
to west, marking one of the highest seismic hazard zones of the island (see location of the epicenters on
the seismic hazard map completed by Frankel et al. in 2011 in Fig. A1 in the annex, as well as on the
topographic map shown in Fig. 1).
For the 2010 event, Calais et al. (2010) and Symithe et al. (2013) showed that this earthquake was caused
by the oblique rupture of a formerly unknown fault dipping towards the north and located immediately
in the north of the EPGF. Data provided by the earthquake.usgs.gov webpage (considering the provided
moment tensor solution; see also Okuwaki and Fan, 2022) indicate that the situation could be similar for
the 2021 event, with a ruptured fault segment dipping towards the north, and mostly located in the north
of the EPGF. Thus, also the recently ruptured fault segment would not belong to the EPGF (which is
essentially a left-lateral strike-slip fault). It could be related to an adjacent blind fault segment with
oblique slip character (left-lateral strike-slip combined with reverse movement) according to the
information available on earthquake.usgs.gov, and to Okuwaki and Fan (2022). The latter further indicate
that especially the eastern part of the ruptured fault showed a more reverse while the western part a
preferential strike-slip mechanism. However, by now there is still no clear answer to the question related
to the fault itself. Therefore, below we will use the term of the 'EPGF zone' that includes the main strike-
slip fault and annexed oblique (or combined) slip fault segments (the two that are now known, i.e., the
one ruptured in 2010 and the one that produced the last earthquake) to denominate the tectonic structure
that produced those two events.
Even though the magnitude of the 2021 earthquake is slightly larger than the one of 2010 (Mw=7.2 and
Mw=7.0, respectively, see information on the earthquake.usgs.gov webpage and by Stein et al., 2021),
the recent event was far less catastrophic as it hit a less populated area compared to the 2010 earthquake
that occurred just near the western entrance of the capital of Haiti, Port-au-Prince. The 2021 earthquake
accounts for about 2250 fatalities (2/3 of which occurred in the provincial city of Les Cayes, located in
Fig. 1), while the 2010 death toll is up to 300,000. However, it quickly became clear that the last event





caused widespread slope failures that could be more intense than in 2010. Therefore, members of our
research team completed some ground control during a one-week field visit along segments of important
roads hit by rock falls near the epicentral region. Additionally, we mapped landslides over the whole area
potentially hit by the 2021 event by using satellite imagery of variable resolution, as it will be explained
in section 2. The main target of this mapping task was to produce an input data set for an extensive
landslide susceptibility analysis that will be presented in an upcoming publication.
Such event-based seismically induced landslide inventories allow us to complete a more systematic
analysis of global patterns of those mass movements, such as size-frequency relationships (Malamud et
al., 2004; Tanyas et al., 2019b), estimates of the expected number of landslides and affected area
(Havenith et al., 2016; Keefer and Wilson, 1989; Marc et al., 2017), and very general earthquake-
triggered landslide susceptibility markers (Tanyas et al., 2019a). At regional scale, event-based landslide
inventories are valuable to understand more specific patterns of seismic slope instability, particularly
with respect to the earthquake mechanism and the geological and climatic context (Gorum et al., 2011;
Tanyas et al., 2022). Below, we will also present statistical characteristics of this new 2021 inventory
compared with equivalent results obtained for the 2010 USGS landslide catalogue published by Harp et
al. (2016); some statistical data are also compared with those of the other inventory completed by
Martinez et al. (2021, USGS Open File report) for the 2021 event and of two additional catalogues
compiled for the 2010 event (by Gorum et al., 2013 and Xu et al., 2014).
Finally, we also mapped landslides existing before the 2021 earthquake by using high-resolution (<=1
m) imagery available on Google Earth Pro®, starting from October 2014 until the end of 2017, to study
some preconditioning of slope instability that was induced in 2021. In particular, it is known that the
region is often affected by hurricanes – the last catastrophic one, 'Matthew' or *'Mathieu'* in French, had
impacted the target area in October 2016. Also, just two days after the main shock, on August 16, another
Hurricane, 'Grace', hit the area and hampered help convoys to reach the areas most impacted by the
earthquake. Right after this event, it was not immediately clear if Grace had contributed to landslide
activity or not; this question will be analyzed in the following sections by comparing landslide
distributions with monthly precipitation maps produced by the 'Global Precipitation Measurement'
(GPM) Mission (NASA) for different periods.



108 Fig. 1 presents an overview map with outlines of landslides mapped by Harp et al. (2016) (shown by

109 light violet - pink polygons, near the 2010 M=7 epicenter), and the recently mapped landslides triggered

110 in August 2021 (outlined in dark red, mainly in the west and south of the 2021 epicenter). This map also

111 shows the approximate paths of the two aforementioned hurricanes near Haiti. Other digital outlines (also

112 those presented in the following figures that also present more detailed views with more clearly visible

113 outlines), such as roads, rivers, faults and coastline, were provided by the Centre National de

114 l'Information Géo-Spatiale (CNIGS) of Haiti.

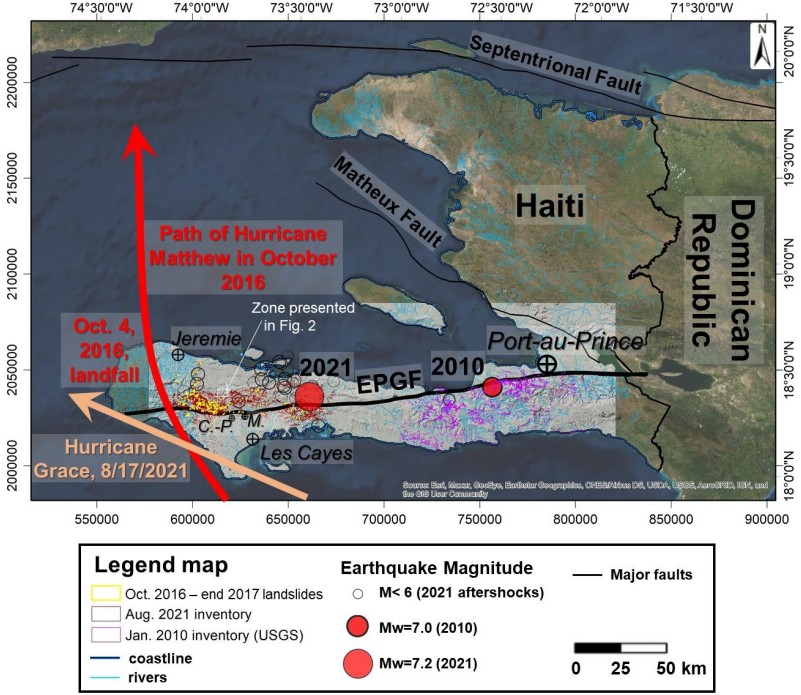

115

116 **Figure 1: Location of the study region in Haiti: Satellite image view of Haiti (by © ESRI), with study region**

117 **highlighted by the hillshade. See also location of the 2010 and 2021 epicenters, and the major cities (see also**

118 **'C.-P.' for Camp-Perrin and 'M.' for Maniche) by hit by those two events. Outlines of major faults are shown**

119 **as well as the indication of the approximate paths of Hurricane Matthew in October 2016 and of Hurricane**

120 **Grace in August 2021. Landslides mapped by Harp et al. (2016) are shown by light violet polygons, and**

121 **recently mapped landslides triggered in August 2021 are outlined in dark red See also location of the zone**

122 **presented in Fig. 2.**




The following sections will provide more detail about the landslide mapping itself, the completion of
landslide statistics, the collection of climatic data and the computation of seismic intensity maps. All
those inputs will be used to explain both the common and the different main markers of landslide
catalogues, respectively, for the 2010 and the 2021 events.
**2    Methodological aspects of landslide and seismic trigger factor mapping**
**2.1 Landslide mapping**
2.1.1 Field observations
Right after the main shock that hit Haiti on August 14, 2021(precisely at 12:29:08 UTC, about 8:30 am
local time), it became clear that many landslides were triggered by this earthquake. Within a few hours
after the main shock, there were reports about rock falls cutting the main road RN7 connecting the large
provincial cities of Les Cayes in the south and Jeremy in the north. Therefore, local members of our
research team checked the situation to support local administration with cleaning the roads. Photographs
of rock falls in the central part of the target area are shown in Fig. 2 (those shown below all occurred in
limestone rocks), together with the locations of the affected sites on a map.

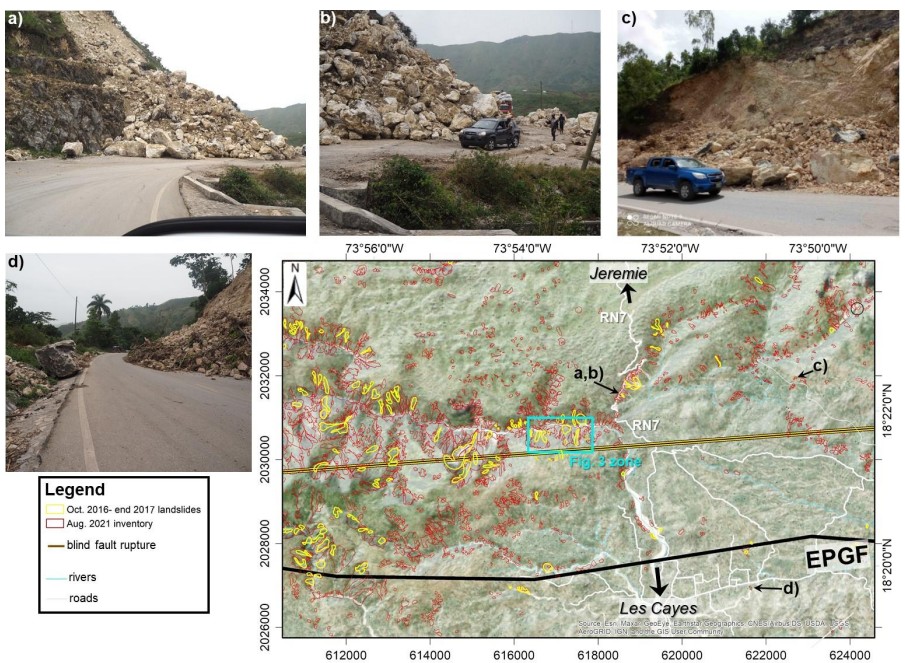

**Figure 2: Examples of landslides photographed in the field, especially along the national road RN7, connecting the two provincial cities of Les Cayes in the south and Jeremy in the north. The map (semi-transparent hillshade on high-resolution satellite imagery, by © ESRI) also shows the different ground failure effects mapped before (yellow polygons) and after the earthquake (dark red polygons). See blue rectangle marking the outline of the view extent shown in Fig. 3 (and in Fig. A2).**

These rock falls were typically not very large (with a volume of generally less than 20,000 m$^3$), but there were many of them and in some cases, it took several days before the street could be reopened. For that reason, we started to detect and map all landslides caused by the earthquake. In addition, during field visits in August 2021, just after the main shock, our teams could confirm that this earthquake had triggered more extensive slope failures (covering wider surface areas) than the previous M=7.0 event in January 2010.

2.1.2 Regional mapping of landslides using remote imagery of August – November 2021

Mapping of earthquake-induced landslides is often done from pre- and post-event optical and radar


satellite imagery, both publicly accessible, like Sentinel-2 (Tanyas et al., 2022) or Landsat-7 and -8 data,
with resolutions starting from 10 m, or commercial higher resolution data that is often made publicly
available for disaster response, through Google Earth, with resolutions of down to 0.5 m e.g. (Harp et al.,
2016; Kargel et al., 2016; Wartman et al., 2013). Sometimes mapping is also supported by (pre-event)
digital elevation data (Gorum eta 1., 2011; Kargel et al., 2016) or even by field or helicopter
reconnaissance. Landslides are mapped at different levels of spatial discretization, e.g. as landslide
initiation points (Gorum et al., 2011), centroid points (Wartman et al., 2013), or landslide polygons
(Tanyas et al., 2022), and with a varying degree of detail, e.g. regarding the minimum mapped landslide
size or the identification of landslide types. The quality and accuracy of the inventories depend typically
on the resolution of the satellite data, cloud cover, and the availability of suitable pre-event data for a
clear identification of co-seismic landslides. A recent review of earthquake-induced landslide inventories
was presented by Tanyas et al. (2017).
In our case, medium-resolution imagery available from the Copernicus Open-Access Hub was used for
the landslide mapping over the whole potentially affected area: Sentinel-2, with 10-m spatial resolution
bands B2 (490 nm), B3 (560 nm), B4 (665 nm) and B8 (842 nm) collected for eight different dates, every
five to six days, between August 14, 2021 (the first one was available about two hours after the main
shock), and the end of September 2021 (an example of a Sentinel 2 image view of this period is shown
in the annex, in Fig. A2a, presenting a view of the zone located in the map in Fig. 2). Analyzing all
images was necessary due to the extensive (but spatially variable) cloud cover present on each image.
Considering that only this medium-resolution imagery was freely available in the beginning, the authors
are aware that the landslides could not be mapped with the highest precision, and that not all smaller
landslides could be identified. However, during the following months, also higher resolution (0.5-1 m)
imagery became available on Google Earth Pro® (GEPro) for about 50% of the potentially affected
region (before December 2021). For these areas, the initial landslide outlines could be refined, and also
smaller slope failures could be mapped; an example of the 'resolution' effect on landslide mapping is
shown in the annex, by Fig. A2, comparing the aforementioned Sentinel 2 image (black-white, projected
on the topography in GEPro) with a higher resolution image of the same landslide zone that became
available on GEPro in September 2021. On the basis of such comparisons between higher and lower
resolution imagery, we could see that most larger landslides are actually composed of multiple initially


smaller and narrower slides and flows, which had coalesced to form a larger coherent mass (while also
on the higher resolution imagery no clear separation could be outlined within this landslide area); actually,
the refinement could only help identify distinct sources of those larger mass movements, but the outline
of the main sliding mass often remained the same. Furthermore, for most landslide zones, no clear
distinction could be made between landslide scarp and deposits, as it can often be observed for such kind
of disrupted mass movements.

2.1.3 Regional mapping of landslides using remote imagery of November 2014 – August 2021, with
focus on pre-seismic changes that occurred in October 2016
For the entire area, also a comparison with pre-event imagery was completed to be sure that only 'co-
seismic' (or nearly co-seismic – see explanation below) slope failures had been mapped; this check was
especially necessary for the identification of the smaller co-seismic landslides. Therefore, the impacted
region was screened by using high resolution (0.5-1 m) imagery available on GEPro for the period
between 2014 and August 2021. A pre-earthquake image (of November 28, 2014) of the same landslide-
impacted area is shown in Fig. 3a, highlighting the contrast between the vegetated slopes present in the
target region and the extensive denudation that occurred during the earthquake of August 2021 (see
images shown in Fig. 3d, identical to the one shown in the annex in Fig. A2a). However, we could also
observe by comparing multiple images available for the pre-event period that some denudation had
already appeared for smaller zones before 2021. Zones marked by narrow debris slides and flows could
be outlined especially on images available for the time just after October 10, 2016. Fig. 3b presents an
image of October 12, 2016 that shows the 'freshest' type of denudation since 2014 (see yellow polygons
outlining such denudation zones), some of which disappeared after a few years (see Fig. 2c), due to
revegetation of the slopes (rapid revegetation can be observed as the whole area is located in tropical
regions). This image and others available for the same period were added to GEPro after Hurricane
Matthew had impacted, in early October 2016, the same area as the one hit by the 2021 earthquake. The
consequences of this 'double' impact on the target region will be analyzed in the sections 3 and 4 on the
basis of precipitation distribution maps.
Actually, Haiti is quite often (at least once per year) crossed by hurricanes or severe tropical storms, some
of which can trigger slope failures over wide areas. One such tropical storm that later developed into the



hurricane called Grace had also crossed southern Haiti, just two to three days after the August 14, 2021,
main shock. We introduce this fact here in the methodological part as it had two consequences for the
landslide mapping: first, right after the earthquake wide areas were covered by clouds during several
days (some higher mountain parts even for weeks); thus, multiple satellite images of different dates (both
Sentinel-2 and higher resolution imagery on GEPro had to be inspected to map landslides over the whole
area. Second, we had to consider that Grace might also have induced slope failures and that landslides
mapped by using post-hurricane imagery were not all seismically triggered, or were at least enlarged by
the effects of Grace. Therefore, by comparing the post-seismic, August 14, Sentinel-2 image (collected
before the Hurricane Grace event) with the one of August 29, 2021 (post-seismic and post-hurricane),
we checked if additional or enlarged slope failures had appeared on the latter. An example of such a
comparison is presented in Fig. A3, where red arrows point to zones marked larger slope failures on the
Sentinel-2 image of August 29, 2021, which were thus most likely reactivated by rainfall during the
Grace climatic event (disregarding here the possible additional influence of aftershocks occurring at the
same time in the region). Unfortunately, due to the extensive cloud cover in mid-August 2021, such a
comparison could only be completed for about 10% of the seismically impacted area. For those cloud-
free zones, we estimate that Grace had mainly induced a widening of the initially seismically triggered
slope failures, but the importance of this reactivation process cannot be quantified due to the extensive
cloud cover and related shadow effects on the surface.




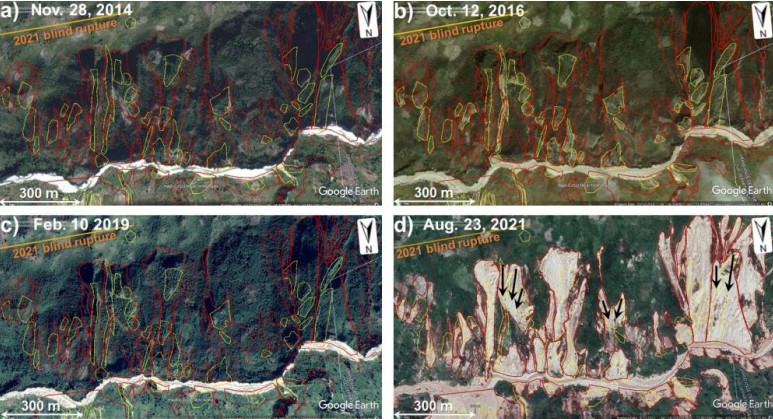


**Figure 3: Evolution of landslides within the zone marked in Fig. 2 between November 2014 (view a), after**

**Hurricane Matthew (Oct. 2016, b), a view of the area of Feb. 2019 (c) and of Oct. 2021, about two months of**
**the earthquake (an image of August exists, but it is partly cloudy). Landslides that occurred during or directly**
**after the Hurricane Matthew event are outlined in yellow and those that were triggered by the August 14,**
**2021, earthquake are shown by dark red polygons; see also black arrows marking the coalescence of landslides**
**with distinct sources (only for three examples shown). All views on © Google Earth Pro.**

**2.2 Landslide distribution statistics, climatic context and a first size-frequency analysis**
In sub-section 3.2, observed total landslide numbers and surface areas as well as other parameters
characterizing the statistics of the two inventories, the new one presented here for 2021 and the one for
2010 by Harp et al. (2016), are compared with 'predicted' ones. The latter numbers are computed
according to prediction laws proposed by Havenith et al. (2016) and Malamud et al. (2004). To estimate
the total number ($N_{LT}$, see Eq. 1) of landslides triggered by a specific earthquake, Havenith et al. (2016)
recommend to take into consideration the shaking intensity factor, (I, based on the Arias Intensity, see
Arias (1970), and thus on the earthquake magnitude, M; see Eq. 6b in the next sub-section), the fault
factor F (depending on the type, FT, and size of the fault rupture, considering also the influence of a
possible surface rupture), the topographic energy (TE, using mainly as parameter the maximum altitude
difference in the affected region), the climatic background (CB) conditions, and the lithological factor
(LF, depending on the presence of soft soils for instance). Related factor values used for the calculations


are compared with estimated minimum and maximum values in Table 2, in the following section.
$N_{LT} = 1000 \times I \times F \times TE \times CB \times LF$ ,                    (1)
Compared with the prediction of the total number of landslides triggered by a specific earthquake
proposed by Havenith et al. (2016), the one recommended by Malamud et al. (2004) is much simpler (Eq.
2) and only based on the earthquake magnitude, M.
$N_{LT} = 10^{(1.29M - 5.65)}$ ,                    (2)
For the calculation of the total area potentially affected by landslides ($A_{Lext}$, area within the maximum
extent of landslide occurrence, equivalent to the area of distribution defined by Marc et al., 2017, and
Tanyas and Lombardo, 2019) Havenith et al. (2016) propose the following Eq. (3), which also directly
considers the earthquake magnitude, M, and the hypocentral Depth, D:
$A_{Lext} = I \times FT \times TE \times CB \times LF \times M \times D^2$ ,                    (3)
As Havenith et al. (2016), Keefer and Wilson (1989) also propose an equation to estimate the total area
potentially affected by landslides during one earthquake event. Their estimate of $A_{Lext}$ is purely based on
the earthquake magnitude, similar to Eq. (2) proposed by Malamud et al. (2004) to estimate $N_{LT}$ :
$A_{Lext} = 10^{(M - 3.46)}$ ,                    (4)
Malamud et al. (2004) do not propose any formula to estimate the total area potentially affected by
landslides during an earthquake event as Havenith et al. (2016) (see Eq. 3), but recommend the following
prediction law (Eq. 5) to estimate the total area effectively covered by co-seismic landslides, $A_{LT}$, based
on the observed or predicted (using Eq. 2, or any other related prediction law, such as the one in Eq. 1)
total number of landslides:
$A_{LT} = 0.00307 \, N_{LT}$ ,                    (5)
All the previous equations were used to compute the respective values presented in Table 1 in sub-section

275    3.2.

Size-frequency relations were computed for the 7091 landslide outlines in terms of frequency-density
function (FDF) on the basis of the measured surface areas, $f(A_L)$. The same statistics were also computed
for the 23,567 landslides mapped by Harp et al. (2016). Therefore, we used the method introduced by



Malamud et al. (2004) for surface areas (Eq. 6):
$$f(A_L) = \frac{\delta N_L}{\delta A_L} \qquad\qquad\qquad\qquad (6)$$
where $\delta N_L$ is the number of landslides with areas between $A_L$ and $A_L + \delta A_L$ (representing the difference
between two landslide surface area classes). Surface areas were calculated in $km^2$. Related distributions
computed, respectively, for each landslide catalogue (for the 2010 one by Harp et al., 2016; and for the
new 2021 inventory) are then compared with theoretical frequency-density distributions, as proposed by
Malamud et al. (2004). The latter are based on the three-parameter inverse-gamma probability
distribution (see equation 3 in Malamud et al., 2004) that is multiplied by the total number of landslides
of simulated events (100, 1000, etc.). In this regard, it should be noted that the original technique
proposed by Malamud et al. (2004) to complete the size-frequency statistics is based on the probability-
density values, corresponding to the frequency-density values divided by the total number of mapped
landslides, $N_{LT}$ (which can be fit by the aforementioned three-parameter inverse-gamma probability
distribution). However, as indicated above, due to the limited amount of high-resolution imagery
available for the area potentially affected by seismic shaking in August 2021, not all small landslides
could be mapped; therefore, the total number of landslides seismically triggered in August, $N_{LT}$, is likely
to be higher than 7091 (even if the potential 'hurricane-effect' is removed, as explained below), and the
probability-density function cannot be correctly computed. For such cases, Malamud et al. (2004)
recommend the computation of the frequency-density function to assess the completeness of the
inventory by comparison with the aforementioned predefined theoretical frequency-density functions, as
it will be shown for the 2010 and 2021 inventories in the following results section.
To provide information about the climatic context covering different periods of time, we used the Global
Precipitation    Measurement    Mission    (GPM,    by    NASA)    data    obtained    via    the
https://giovanni.gsfc.nasa.gov/    website,    corresponding    to    the    merged    satellite-gauge    monthly
precipitation estimate (in mm), assessed with a resolution of 0.1°. Related maps were extracted for all
months between August 2000 and July 2021, and also for the specific months of October 2016 and August
2021, as well as for all October months between 2000 and 2020. Note, we also extracted maps for shorter
periods around the climatic events of Matthew in 2016 and Grace in 2021, but those did not provide any
additional information. Additionally, we tried to support these merged satellite - rain gauge estimates by





additional ground measurement data. However, the *Centre National de l'Information Géo-Spatiale,*
*CNIGS*, of Haiti, informed us that such data would not be available; therefore, we can only rely on these
regional estimates.
**2.3 Mapping of seismic landslide triggering factors**
The aforementioned climatic data are supposed to help us better understand the pre-conditioning of slope
stability in the target area and thus will also be used below for the interpretation of the landslide
distribution statistics. However, it is obvious that for such an event the main trigger factors are still related
to earthquake shaking; those have to be assessed to understand why extensive slope instability could be
observed in one zone and only isolated minor failures occurred in another one. Such an analysis is
completed both for the 2010 and 2021 events, by computing the Arias Intensity distribution maps (for
2010, comparing the results with the landslide distribution as observed by Harp et al., 2016).
The Arias Intensity, Ia, can be considered as a quantitative measure of the degree of shaking (in m/s) on
the surface. With respect to any other intensity characterization (including the one based on surveys) it
has the advantage to be more objective and comparable for different earthquakes (according to Harp and
Wilson, 1995). Wilson and Keefer (1985) were the first to try to correlate seismically triggered landslide
distributions with this intensity measure. They also defined the following empirical attenuation
relationship (Eq. 7a) in terms of magnitude (M) and hypocentral distance (R):
$$\log(Ia) = -4.1 + M - 2\log(R) + 0.5P \; , \tag{7a}$$
where P considers a possible deviation from the main law (P=0 stands for the average value).
Afterwards, Keefer and Wilson (1989) have reviewed the application of this formula and defined a new
one (Eq. 7b), for magnitudes greater than 7:
$$\log(Ia) = -2.35 + 0.75M - 2\log(R) \; , \tag{7b}$$
We applied the latter equation as both the 2010 and 2021 can be considered as M>=7 events. The R-
value represents the hypocentral distance map, here computed by using as source zone the blind fault
rupture segments of the 2010 and 2021 events (with 0 km epicentral distance and 10 km hypocentral
depth along the respective segment; information extracted from earthquake.usgs.gov).
All equations introduced above have been applied to obtain the computation results presented below, in



the sub-sections 3.2 and 3.4.

### 3    Results: landslide inventory statistics and analysis of trigger conditions

This section first summarizes a series of landslide type and general distribution characteristics. Second,
landslide inventory and size-frequency statistics are presented and supported by an inventory
completeness analysis. Third, a study of possible climatic slope failure preconditioning and post-seismic
landslide surface changes is presented, which also compares landslide distributions with monthly
precipitation maps (using output maps of the Global Precipitation Measurement Mission, GPM, produced
by the NASA, for different periods, according to Acker and Leptough, 2007). Fourth, the landslide
occurrence observed in 2010 and in 2021 is compared with respective shaking intensity prediction maps.

### 3.1 Landslide type and distribution characteristics

Before analyzing specific statistical values of the two landslide inventories, the one compiled by Harp et
al. (2016) for the 2010 event and ours completed after the August 2021 earthquake, we first have a look
at the general respective spatial landslide distributions and provide basic information on the type of the
mapped landslides.
The map presented in Fig. 4a shows that the global extent of landslides triggered in 2010 (pink outlines
within the pink maximum extent polygon) and in 2021 (dark red outlines within the dark red maximum
extent polygon) is quite similar (exact values are presented in Table 1). This map also show the location of
the main shock and aftershocks (empty circles, from earthquake.usgs.gov) and the outline of the (roughly 80 km
long)        blind        fault        rupture        (extracted        from        USGS        page:
https://earthquake.usgs.gov/earthquakes/eventpage/us6000f65h/finite-fault). Outlines of zones shown in Fig. 5 are
shown by light blue rectangles. A major difference between the two landslide distributions can mainly be
observed with respect to the location of the EPGF zone. While most landslides occurred in the south of
the fault zone in 2010, a relatively symmetric distribution of landslides with respect to the location of the
EPGF zone can be observed for the 2021 event. This is mainly due to the fact that the fault segment that
ruptured near EPGF in 2010 is located close to the coast (actually just in the south of the coast, as can be
seen in the map in Fig. 4a), and thus only limited onshore surface areas could be affected by landslides





in the north of the EPGF zone in 2010, while the location of the fault segment that ruptured in 2021 is
more central within the southwestern peninsula of Haiti (see focus on this region in Fig. 4b).
Another important observation is that there seems to be a gap between the zone affected by landslides in
2010 and the one affected in 2021. This means that, according to our present observations, the 2021
earthquake did not reactivate landslides triggered in 2010 – due to the large distance (> 60 km) between
the fault ruptures. However, it should be noted that this check could only be completed so far with the
10-m resolution Sentinel-2 imagery. Now, we cannot exclude that very small landslides (that we cannot
identify on Sentinel 2 imagery) triggered in 2010 had been reactivated in 2021.

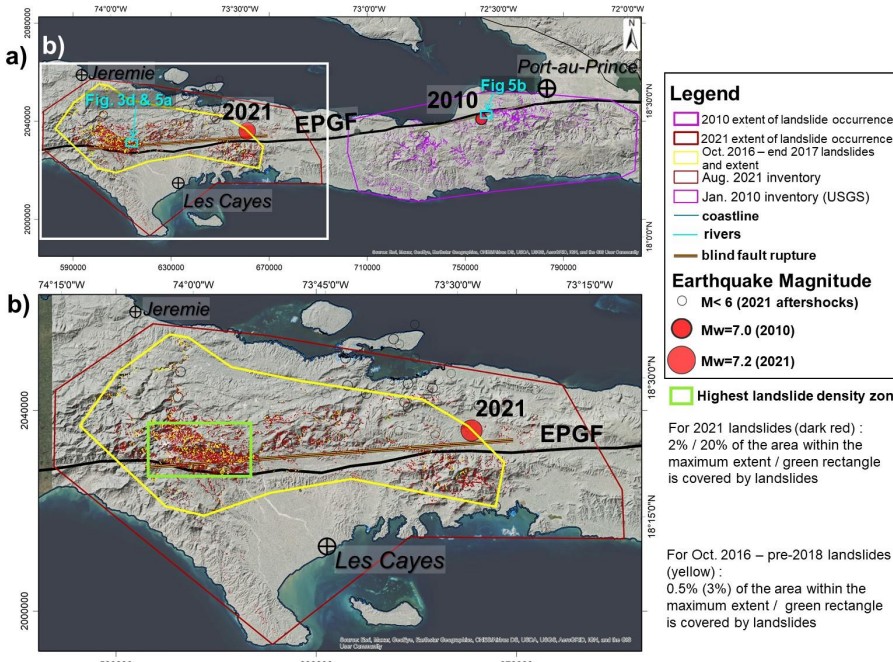


**Figure 4: a) Study region with areas affected, respectively, by the two Mw >= 7 events. Individual landslides**
**triggered in 2010 (Harp et al., 2016, inventory) and our landslides mapped for the 2021 earthquake and**
**October 2016 hurricane events are shown, respectively, as pink, dark red and yellow polygons. The maximum**
**extent of landslides triggered in 2010, in 2021 and 2016 is outlined, respectively, by the large pink, dark red,**
**and yellow polygons. b) Focus on the region hit by the August 2021 earthquake, with 7091 landslide locations.**
**Map background by © ESRI.**




An important consequence of the specific location of the ruptured fault segments is that a few dozens of
landslides with a surface area larger than 2000 $m^2$ had occurred along the shore in 2010, where the two
or three largest ones (likely including an important submarine part) had massively impacted the ocean
and, thus, had produced up to 3 m-high tsunami waves (see Olson et al., 2011; Poupardin et al., 2020;
Fritz et al., 2013; Sassa and Takagawa, 2018) while there is not a single report of a major coastal landslide
for the 2021 event – as the fault rupture occurred at a distance of minimum 10 km away from the nearest
shoreline. Instead, a wider onshore area was exposed to high intensity earthquake shaking during the
2021 event. The related impact will be analyzed below on the basis of the statistical values presented in
Table 1.
Concerning the types of landslides triggered by the 2021 earthquake, we can say that by far most of them
can be classified as debris slides or flows (see examples in the GEPro view presented in Fig. 5a) and as
medium-size (most with a volume of less than 20,000 $m^3$) rockfalls (as shown above in Fig. 2). Thus, we
estimate that at least 95% of all landslides mapped are relatively shallow (with a depth of less than 10
m). Actually, not a single large massive landslide ($> 10^7$ $m^3$) could be identified. A similar observation
was made by Harp et al. (2016) for the landslides triggered in 2010 (see view in Fig. 5b). However, when
comparing individual landslides induced in 2021 with those triggered in 2010, the latter are almost
systematically narrower than those of 2021 (compare the very narrow slides and flows in Fig. 5b with
the typically wider ones in Fig. 5a), while located in similar geological (limestone) and topographic
(hilly-mountainous) environments. Actually, in the so-called *Ravine du Sud* (Gorge of the South), part of
which is shown above in Fig. 3 (and in the annex, in Figs. A2 and A3), numerous very extensive slope
failures (but still relatively shallow) could be observed; most of them formed by coalescent neighboring
debris slides. Thus, entire slope units (delimited by upper and lateral slope crests and the valley bottom)
finally collapsed as one single mass movement. Such kind of extensive slope failures occurred far less
frequently in 2010 – at least onshore, while at least a few aforementioned coastal and mostly submarine
landslides must have been quite massive as their impact had triggered tsunami waves, as indicated above.
This assumption cannot be further verified as no higher resolution marine floor surface data are available.
However, we are aware that a full mapping of submarine or mixed subaerial-submarine slope failures



would be necessary to better understand the landslide distribution characteristics, especially for the 2010
event, as further discussed below.
The fact that no really massive landslides had occurred (onshore), both in 2010 and 2021, also explains
why only a few longer lasting landslide dams had formed on the rivers. We could identify only about 100
minor dams (with a volume of less than 50,000 $m^3$, according to our estimate, based on the maximum
surface area value of about 5000 $m^2$ measured for the largest observed dam and a related maximum
possible thickness of 15 m, compared with the local morphological setting) after the August 2021 main
shock, most of which had disappeared before the end of October 2021; and, only a few dozens of them
were impounding temporary lakes. In this regard it should be noted that Martinez et al. (2021), who had
also mapped landslides triggered by the 2021 Nippes earthquake (4893, according to their open file
report), have outlined almost 300 (at least partial) landslide dams after the event. However, they also
indicate that most of them failed a few days after formation; still, at the time of publication of their open
file report in December 2021, they consider 35 of the remaining dams as potentially hazardous. Here, we
will not further analyze this aspect as any related hazard assessment would require a site-specific
approach that is not targeted by this first study completed at regional scale.

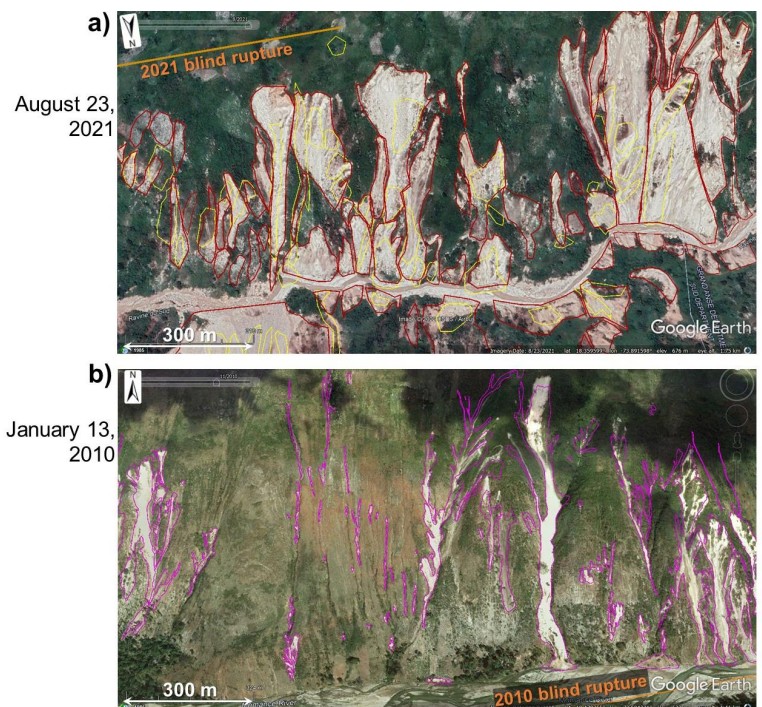


**Figure 5: a) GEPro view (© Google Earth Pro) of landslides triggered by the August 2021 earthquake. b)**

**GEPro view (© Google Earth Pro) of landslides induced by the January 2010 main shock (with landslide**

**outlines by Harp et al., 2016).**

While debris slides are the predominant type of 2021 slope failures in the central mountain ranges,

widespread soil slides (but of smaller volume, typically of less than 10,000 m$^3$) had occurred along the

hills (with an absolute crest altitude of less than 400 m, and a relative height of less than 200 m with

respect to the nearby valley bottom) of the peninsula located in the southwest of Les Cayes (southern

part of map in Fig. 4b). As the slopes are very gentle, often seem to be less than 5°, it could be that those

failures, many of which affected agricultural areas (marked by brownish disrupted fields), are related to

liquefaction phenomena. However, also this observation has to be reexamined, by ground-control and

site-specific studies, as the remote analysis based on 1-m resolution imagery does not allow us to fully

confirm this interpretation.





**3.2 Landslide inventory and size-frequency statistics**

3.2.1 Landslide inventory statistics

Table 1 presents an overview of general landslide inventory statistics, for both the 2010 and 2021 events. The numbers in the first row show that apparently fewer landslides have been triggered in August 2021 (considering also the number of 4893 landslides published in the open file report by Martinez et al., 2021) than in January 2010. At least two inventories, the one by Harp et al. (2016) and the one of Xu et al. (2014), include far more landslide outlines (23,567 for the first, 30,828 for the second) than our catalogue for 2021 (7091). Only the inventory by Gorum et al. (2013) that was the first one to be completed for the 2010 event contains fewer data (4490 points – not polygons - marking the landslide location). However, paradoxically, a much wider surface area is covered by the apparently fewer 2021 landslides (a total area of 84 km$^2$, see row 3 in Table 1) than by the more numerous 2010 landslides (sum of surface areas of about 25 km$^2$, calculated for the Harp et al., 2016, inventory). This discrepancy will be discussed below, considering the fact that 2021 landslides could only be mapped from higher resolution imagery for about half of the potentially affected area (in the eastern part). For the western zone, only Sentinel-2 images were available until the end of 2021. Those 10-m resolution images typically do not allow for the (complete) mapping of landslides smaller than 2000 – 3000 m$^2$. Therefore, we focus only general landslide inventory statistics, first, by comparing the observed landslide numbers with those predicted by Havenith et al. (2016) and Malamud et al. (2014), respectively, for the two earthquakes – always keeping in mind that the 2021 inventory is not complete for landslides smaller than about 3000 m$^2$ (this value will also be analyzed below on the basis of the size-frequency relationship). As introduced above (see Eq. 1), according to Havenith et al. (2016), this number depends on the seismic intensity (I, using as input the Ia value computed for the respective earthquake magnitude), the fault factor (type, size and possible surface rupture), the topographic energy (maximum difference of altitudes in the affected area), the climatic background (in this case marked by tropical wet climate), and the lithological factor (here using an average type, for rocks in general). For the precise classification of the different factors, the reader is referred to Table 1 in Havenith et al. (2016). Here, we used the values presented below in Table 2 (considering both events in 2010 and 2021), which indicate the following:

1) the shaking intensity values, I=0.74, in 2010, and I=1 in 2021 are characteristic for the respective





magnitudes (note, this factor can reach a value of up to 3.5 in the case of high-magnitude earthquakes,
with Mw > 8);
2) the fault factor, F=2.25, can be considered as similar in both cases, marked by an oblique slip that
occurred along a fault segment with a length of 50-100 km, with no clear surface rupture (note, F can
reach a value of up to 6 in the case of a surface rupture of an activated reverse fault segment with a length
of more than 300 km, such as observed for the Wenchuan earthquake in 2008);
3) the topographic energy value, TE=2, in both cases characterizes a surface morphology marked by local
altitude changes of more than 500 m within a hilly region (only smaller mountains, with an altitude of
less than 2500 m can be found in the regions affected by the 2010 and 2021 events; note, Havenith et al.,
2016, selected a value 4 to mark the high steep slopes in the Longmenshan Mountains affected by the
Wenchuan earthquake in 2008);
4) the climatic background factor, CB=1.5 marks relatively wet conditions for the 2021 event while CB=1
indicates average conditions for the 2010 event (the higher value chosen for 2021 considers some
preconditioning of slope instability by Hurricane Matthew, as explained in the next section; note,
Havenith et al., 2016, selected a value CB=2 for the very wet conditions that can be found in the
Longmenshan Mountains affected by the Wenchuan earthquake, characterized by yearly precipitation
values of more than 3000 mm – while the target areas in Haiti are marked by values of about 2000 mm);
5) the lithological factor, LF=2, indicates that both weathered rocks and soft soils can be found in the
affected area (note, Havenith et al., 2016, selected a maximum value, LF=4, for the Haiyuan-Gansu-
Ningxia earthquake event, China, in 1920, as it affected an area that is almost entirely covered by Loess
deposits, which are highly susceptible to slope failure).
When these different factor values are combined according to Eq. (1) presented above, the total numbers
of landslides, $N_{LT}$, predicted for the 2010 and 2021 events are, respectively, 6694 and 13,476. These
values can be compared with the numbers predicted by the simple equation (Eq. 2), proposed by
Malamud et al. (2004), using only the earthquake magnitude as input: 2399 for the 2010 event and 4345
for the 2021 event. The latter prediction seems to clearly underestimate the observed numbers of
triggered landslides, while those predicted by using Eq. (1) by Havenith et al. (2016) provide intermediate
values: larger than the number observed by Gorum et al. (2013) but smaller than the numbers observed
by Harp et al. (2016) and by Xu et al. (2014). The two predictions (Eq. 1 and 2) were also applied to the



2021 event; the first one producing a higher $N_{LT}$ (=13,476) than the observed value, the second one
producing a lower value (=4345).
As shown on the maps in Fig. 5, also the total area within the maximum extent of landslide occurrence,
$A_{Lext}$, was outlined and then measured for the 2010 and 2021 events. Actually, related areas are relatively
similar: 4400 km$^2$ for 2021 and 4100 km$^2$ for 2010. These values can be compared in Table 1 with the
predictions by Havenith et al. (2016) and by Keefer and Wilson (1989), corresponding, respectively, to
3124 and 3467 km$^2$, for the 2010 event, and to 6470 and 5495 km$^2$, for the 2021 event. In this case, the
very simple equation proposed Keefer and Wilson (1989) provides an estimate of $A_{Lext}$ that is closer to
the observed value than the one produced by the more complex relationship proposed by Havenith et al.

499    (2016).

The third row of Table 1 compares the total observed slope areas affected by landslides, $A_{LT}$,
corresponding, respectively, to a value of 24.86 km$^2$ for the 2010 event and of 84.38 km$^2$ for the 2021
event, with the values predicted by Eq. (5) by Malamud et al. (2004) for each event. For 2010, we applied
this relationship to the three observed values indicated in the first row and by using the $N_{LT}$, predicted
respectively by Havenith et al. (2016) and Malamud et al. (2004). Among all total landslide surface area
values predicted for the 2010 event, it can be seen that the one based on the Havenith et al. (2016) $N_{LT}$
estimate produces the best fit (=20.55 km$^2$) when compared with the observed value of 24.86 km$^2$. For
2021, the respective predictions all underestimate the observed total landslide surface area value, $A_{LT}$, by
a factor of at least two, even when the highest $N_{LT}$ estimate (using Eq. 1) by Havenith et al. (2016) is
used. Using the preceding information, it is also useful to compare the density values (here, expressed
in %) of landslide areas within the maximum extent surface area, which correspond to 0.5% and 2%,
respectively for the 2010 and 2021 events. Within the green rectangle (zone with highest landslide density)
shown in Fig; 4 above, even 20% of all the area is covered by landslides. Possible explanations for the
much larger total area (and the higher density) of landslides triggered in 2021 compared with 2010 will
be provided in the discussion.
The fourth and fifth rows show that the smallest landslide mapped by Harp et al. (2016) has a surface
area of 0.5 m$^2$ and their inventory contains 6587 landslide polygons smaller than 100 m$^2$ while our
inventory for 2021 only includes one landslide with a surface area smaller than this value. This
comparison also confirms that our inventory is likely to be incomplete for such small landslides, as there



is no physical reason why there would be much fewer smaller landslides triggered in 2021 than in 2010.
On the other hand, the largest landslide mapped for the 2021 event (>400,000 m$^2$) has almost twice the
size of the largest one that occurred in 2010, when actually only 2 landslides larger than 100,000 m$^2$ had
been triggered; in 2021, we could outline more than 100 landslides larger than this value. And, for these
larger landslides we can be sure that we mapped them all and outlined them correctly, without
amalgamating distinct slope failures.
Finally, Table 1 provides information about the distribution of the 2010 and 2021 landslides with respect
to the blind fault rupture projected on the surface (near the EPGF outline). As already introduced above,
a much larger number of landslides occurred in the north of the latter in 2021 (=4678) compared to 2010
(=2548, at least for onshore slope failures); consequently, more landslides occurred in 2010 in the south
of the respective blind fault rupture. As the total number of mapped landslides is much larger for the
2010 event (which also means that only the relative proportions should really be considered), the
difference between those numbers is very high: 21,019 occurred in the south of the fault rupture in 2010
(about 90% of all landslides) and only 2420 in the south of the respective fault rupture in 2021 (about
35%). However, when the total surface area affected by landslides is considered, the 2021 event affected
more zones both in the south and the north of the fault rupture than the 2010 event, while the distribution
of landslides for each event with respect to the fault rupture remains the same also when considering the
affected surface areas: they are much larger in the south of the fault rupture for the 2010 event but larger
in the north for the 2021 event. The main explanation for this difference has already been provided above:
the fault segment that ruptured in 2010 is located close to the coast, with limited onshore surface areas
being exposed to landslide activity in the north of the respective fault rupture, while the location of the
fault rupture in 2021 is more central with respect to the shorelines of the southwestern peninsula of Haiti.









**Table 1: 2010 and 2021 landslide inventory characteristics – where not specified for the 2010 event, using the**
**Harp et al. (2016) inventory. The largest values for each specific observation/estimate (if more than 1 indicated)**
**are bold.**

| Landslide inventory parameters/predictions | 2010, Mw=7.0 | 2021, Mw=7.2 |
|---|---|---|
| Observed number of landslides, $N_{LT}$ | >4490[a] / 23,567[b] / **30,828**[c] | **7091**/4893[d] |
| Havenith et al. (2016) $N_{LT}$ prediction 1 | 6694 | 13,476 |
| Malamud et al. (2004) $N_{LT}$ prediction 2 | 2399 | 4345 |
| Area of region potentially affected by landslides, $A_{Lext}$ (km$^2$) | 4100 | 4400 |
| Havenith et al. (2016) $A_{Lext}$ prediction 1 | 3124 | 6470 |
| Keefer and Wilson (1989) $A_{Lext}$ prediction 2 | 3467 | 5495 |
| Total surface area of landslides, $A_{LT2}$ (km$^2$) | 24.86 | 84.38 |
| Malamud et al. (2004) $A_{LT}$ prediction : | | |
| for the observed $N_{LT}$ | 13.8[a] / 72.3[b] / **94.6**[c] | 21.8 |
| for the $N_{LT}$ prediction 1 | 20.55 | 41.4 |
| for the $N_{LT}$ prediction 2 | 7.36 | 13.3 |
| Smallest landslide (m$^2$) | 0.5 | 75 |
| Number of landslides smaller than 100 m$^2$ | 6587 | 1 |
| Largest landslide (m$^2$) | 234,370 | 409,479 |
| Number of landslides larger than 100,000 m$^2$ | 2 | 103 |
| Total number of landslides in the north (N) / south (S) of the fault rupture | N= 2548 / **S= 21,019** | **N= 4678** / S= 2420 |
| Total surface area of landslides in the N / S of the fault rupture (km$^2$) | N= 2.45 / **S= 22.41** | **N= 58.31** / S= 26.07 |

[a] Number of landslides observed by Gorum et al. (2013), [b] by Harp et al. (2016), [c] by Xu et al. (2014),
and [d] by Martinez et al. (2021).

In addition to the numbers shown in Table 1 and explained above, we also provide two values for the
smaller landslide inventory compiled for the period between October 10, 2016 and the end of 2017. For
this period, 625 landslide zones have been outlined (see yellow polygons shown on the views and map
in Figs. 3 and 4), covering a total surface area of 9.5 km$^2$ (located within an area of maximum extent of
these landslides of 1770 km$^2$ as outlined in yellow, above in Fig. 4) This also means that about 0.5% of
the area within the maximum extent was covered by landslides. Highest concentration of landslides can



be observed within the green rectangle shown in Fig. 4, where 3% of the total area is covered by
landslides. However, we must indicate that these values represent approximations as only 50% of the
potentially affected area is covered by cloud-free imagery on GEPro for this period, most of which
actually covers the short period between October 10 and 28, 2016 (just after Hurricane Matthew event).
Post-2017 imagery was not used as we could observe that many landslides identified shortly after Oct.
10, 2016, had already 'disappeared' in 2018-2020 due to revegetation of the affected area (see, above,
the comparison between GEPro views of October 2016 and February 2020 in Fig. 3).
**Table 2: Factors contributing to the total number and surface area of landslides triggered by the 2010 and the**
**2021 earthquakes, according to the prediction proposed by Havenith et al. (2016). The minimum and**
**maximum values proposed by Havenith et al. (2016) are also indicated, the latter with information on the**
**event – region, to which this maximum factor value was attributed.**

| Haiti Events/ Factors | Shaking Intensity, I | Fault Factor, F (type, FT) | Topographic Energy, TE | Climatic Background, CB | Lithological Factor, LF | Hypocentral Depth, D (km) |
|---|---|---|---|---|---|---|
| 2010 | 0.74 | 2.25(1.5) | 2 | 1.5 | 2 | 10 |
| 2021 | 1 | 2.25(1.5) | 2 | 1 | 2 | 10 |
| min. values | 0.1 | 0.75 | 1 | 0.5 | 1 | 10 |
| max. values (event - region) | 3.4 (Chile, 1960) | 6 (Wenchuan, 2008) | 4 (Wenchuan, 2008) | 2 (Wenchuan, 2008) | 4 (Haiyuan-Gansu-Ningxia, 1920) | 226 (Hindu Kush, 2002) |


3.2.2 Landslide size-frequency statistics
We also computed frequency-density values for various landslide surface area classes as shown on the
graph in Fig. 6. There are two important parameters to be analyzed for the observed frequency-density
distributions, according to Malamud et al. (2004), among others: the first part is represented by the
power-law decay (see also introduction in Stark and Hovius, 2001) that appears as a linear decay in the
log-log graph below; the second part is the so-called 'rollover', which can be observed for a landside
surface area where the exponentially decreasing number of larger landslides turns into a decrease of the
number of smaller landslides. Here, we will only focus on the power-law decay that can be observed for
the larger landslides, for which we consider both, the 2010 and the 2021, inventories as complete. Fig. 6
shows that such a power-law decay can be observed for 2010 landslides larger than 2000 m$^2$ and for 2021



landslides larger than about 4000 m$^2$. This comparison confirms the likely incompleteness of the 2021
inventory, even for landslides smaller than 4000 m$^2$. The rollover part will not be analyzed here as it
occurs for smaller landslides, well below this limit of completeness of our 2021 inventory (referring to
estimates by Malamud et al., 2004).
For the larger landslides, the comparison between frequency density outputs of the 2010 and 2021
landslide inventories presented in Fig. 6 first shows that related values are higher for the latter catalogue.
Actually, related frequency density values are three times larger for the landslide size class of 10,000 m$^2$
and even twelve times for the one of 100,000 m$^2$. And, for those larger landslide classes, the absolute
value of the power-law decay is slightly higher (-2.57, for the pink line fitting the 2010 data) for the 2010
inventory than for the 2021 one (-2.03, for the red line fitting the 2021 data). Thus, the relative smaller
decay exponent observed for the 2021 landslide inventory explains why related frequency density values
are increasingly (i.e. for larger landslide sizes) higher compared with the 2010 values observed for the
same landslide size classes. These different size-frequency characteristics of the 2010 and 2021
landslides inventories will shortly be discussed below (considering the constraint of inventory
completeness for both events), but the most important information to be retained at this level is that for
all landslide classes larger than 4000 m$^2$ more landslides have been observed in 2021 than in 2010.

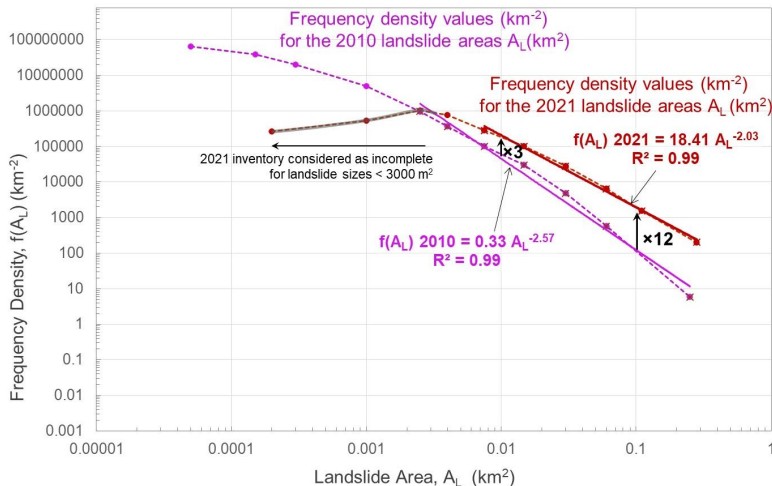


**Figure 6: Frequency density graphs developed for the 2010 (in pink, by Harp et al., 2016) and the new 2021**
**landslide inventories (in dark red), with related power-law decays outlined.**





**3.3 Climatic (pre- and post-seismic) conditioning of slope instability**

The climatic influence on landslide occurrence (in 2021) has been introduced above, by considering the possible impacts of hurricanes on slope failure occurrence, marked both by preconditioning of slope instability and by post-seismic intensification. We first start analyzing the last effect, by considering the potential impact of Hurricane Grace on post-seismic landslide intensification, on August 16-17, 2021 (when it had crossed the target region and was actually classified as tropical depression at that stage). A possible effect of related rainfalls on landslide occurrence has already been highlighted, for instance, on the AGU Landslide blog (by Petley, D., 2021, on blogs.agu.org/landslideblog). This effect could be confirmed when we compared Sentinel-2 imagery collected right after the earthquake (2h after the main shock) with images remotely sensed after August 17, 2021. As indicated above and shown in Fig. 4, an intensification of denudation could indeed be observed after the tropical storm Grace event. However, one important limitation has to be highlighted: this comparison could only be completed for about 10% of the area potentially hit both by the earthquake and by Grace, due to the intense cloud cover present in the target region during that period. Furthermore, another effect could have contributed to slope failure intensification after the main shock on August 14, the one related to the aftershocks (see empty circles shown in all maps above), but analyzing this effect would require a refinement of the satellite image analysis both in space and time, which is hardly possible considering the extensive cloud cover present in the target area when all those seismic shocks occurred. Here, we will focus on the possible climatic influence, which can better be outlined when comparing the landslide distribution with actual precipitation maps. Therefore, we used the aforementioned Global Precipitation Measurement Mission (GPM) data. Fig. 7 presents the three following types of GPM maps: average monthly precipitation maps for the whole period of 2000 – July 2021, for all months of October between 2000-2020, and for October 2016, when Hurricane Matthew had crossed the island.

While comparing the average monthly precipitation rates between 2000 and 2021 (Fig. 7a) with the one of October 2016 (Fig. 7c) we can see that, for the latter month, a peak of intensity of 626 mm can be observed for the area between Gran Rivière De Nappe and Petite-Rivière-de-Nippes, situated immediately in the north of the epicentral area of the 2021 main shock Actually, the whole area potentially affected by the August 2021 earthquake had been exposed to clearly higher precipitation rates





of more than 400 mm in October 2016, while, according to the GPM, average precipitation recorded in
October between 2000 and 2020 varies typically between 200 and 320 mm (as shown by the map, in Fig.
8b). For October 2016, those values were also the highest ones compared with the rest of the country;
this clearly indicates that they must be related to a specific climatic event, which can easily be identified
as Hurricane Matthew that had crossed the western peninsula (including the region hit later by the August
2021 earthquake) on October 4-5, 2016. And, precisely for this region that had been exposed to abnormal
precipitation rates in October 2016, we could outline 625 landslides triggered after the Hurricane
Matthew event, and before the end of 2017 (yellow polygons shown above in the maps in Figs 1, 2, 3
and 4 and below in Fig. 7). And, most of these October 2016 – end of 2017 landslide zones (at least 90%
of them) are located within those mapped for the August 2021 seismic event (which are still marked by
a much higher level of denudation compared to the October 2016 activation). In the discussion, we will
analyze how such Hurricane Matthew might have preconditioned slope instability in the region hit by
the August 14, 2021, earthquake. We will also consider a general influence of tropical storms on the wide
distribution of the landslides triggered in 2010 (and also for those triggered in 2021, in addition to the
Hurricane Matthew effect).

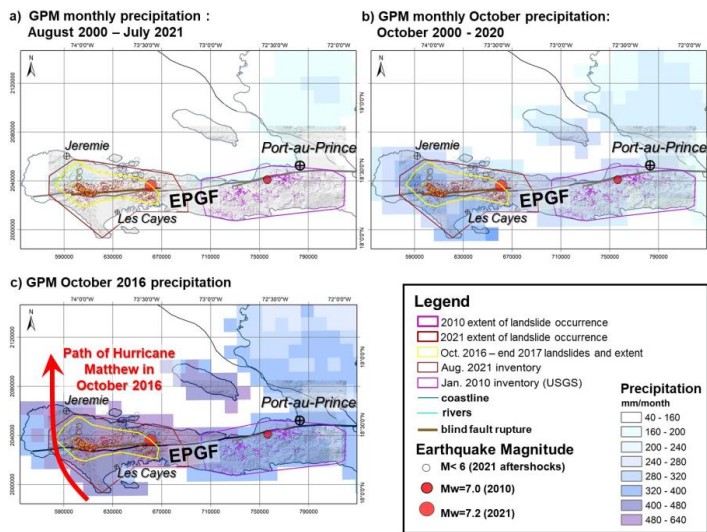


**Figure 7: Monthly © Global Precipitation Measurement Mission (NASA) maps (0.1° resolution, values in**
**mm/month) for southwestern Haiti, (a) for all months between August 2000 and July 2021, (b) for the month**
**of October between 2000 and 2020, and (c) for October 2016 (marked by the Hurricane Matthew event).**





By comparing equivalent data (not shown here) of the merged satellite-gauge precipitation estimate for
August 2021 with the monthly precipitation map averaged for all months of the previous 20 years, we
can clearly see that August 2021 was indeed marked by a higher precipitation rate, which is most likely
related to the Grace event. However, the most intense precipitation did not affect the region hit by the
2021 earthquake but the eastern part of the peninsula, roughly covering the same region as the one
affected by the 2010 event (note, we did not check any landslide reactivation after Grace for that area).
The region hit by the 2021 earthquake was not affected by much higher monthly precipitation rates than
usual: for the central seismically affected zone, in the north of Les Cayes, about 240-280 mm had been
recorded in August 2021, against a monthly average of 200 mm. Thus, just by considering these data,
one would not expect an important climatic contribution to slope failure occurrence in the region affected
by the 2021 earthquake. Still, an influence could be observed and this is likely to be related to the
concentration of most of the 'monthly precipitation' of August 2021 within the two days (Aug. 16 and
17) of the Grace tropical storm event, just two days after the 2021 main shock. As indicated above, we
estimate that related precipitation has resulted in an increase of landslide surface areas of about 10-15%.
Due to the limited extent of zones where this check can be made (only considering the cloud-free areas
on the Sentinel-2 image of August 14, 2021), it was decided to map all areas covered by landslides after
August 14, 2021, also those which are likely to have been (re)activated by rainfall – the total effect of
which can barely be controlled and quantified outside the 10% of cloud-free zones visible on the image
collected right after the main shock. The only 'correction' that can be made is to reduce the total surface
area mapped as landslides by those 10-15% to estimate the one that was actually affected by co-seismic
slope failures: thus, instead of considering the value of 84 km$^2$, it is possible that co-seismic landslides
covered a total surface area of 'only' 75-78 km$^2$ – which is still three times more than the total surface
area covered by 2010 co-seismic landslides (close to 25 km$^2$).

**3.4 Shaking intensity maps**
Above, we fist analyzed the possible climatic influence on seismically induced slope failures as it could
affect the landslide distribution and thus has to be taken into consideration when assessing and
interpreting the seismic effect on landslide occurrence. The latter will only be analyzed here at regional



scale. Therefore, we compare the landslide distributions observed for the 2010 and 2021 events with the
respective estimated Arias Intensity (Ia) attenuation maps, computed by applying Eq. (7b) introduced
above, as recommended by Wilson and Keefer (1985) and also by later studies (e.g., Harp and Wilson,
1995, among many others). The map in Fig. 8a presents the 2010 and 2021 mainshock Ia attenuation
values, with a maximum shaking intensity of 11.2 m/s computed for the 2021 event and 7.9 m/s for 2010
(respective maps are partly overlapping in the central region, but not summed up, keeping the individual
values). This map shows that all 2010 and 2021 landslides are included within a zone marked by an Ia
threshold of 0.2 m/s (close to the one proposed by Keefer and Wilson, 1989, for disrupted slides and
falls). Actually, for 2021, 99% of the total landslide surface areas are even located within a zone marked
by Ia values lager than 1 m/s; however, only 80% of the total surface areas of the 2010 landslides are
included within the respective Ia >= 1 m/s zone. Thus, the latter mass movements appear as more
'dispersed' with respect to the estimated seismic intensity attenuation than the 2021 ones. The latter are
most concentrated, as indicated above, within the green rectangle (see Fig. 8b, marked by Ia values of 4-
11 m/s) with an area of 200 km$^2$ that contains 40 km$^2$ of landslide-covered zones (=20% of total area).
Notwithstanding the relative dispersion of 2010 landslides, and the overlap of Ia values larger than 0.2
m/s in the central zone between the two blind fault ruptures of 2010 and 2021, not a single landslide of
2010 seems to have been reactivated in 2021. This observation raises the question if the central 'landslide
gap' is due to an overestimation of the Ia values in this central zone (as this zone is marked by Ia values
above the aforementioned minimum threshold of 0.2 m/s, for both events, and thus should have been
affected by landslides both in 2010 and 2021, according to the shaking intensity prediction parameter),
or if this zone is simply less susceptible to (seismic) slope failures.


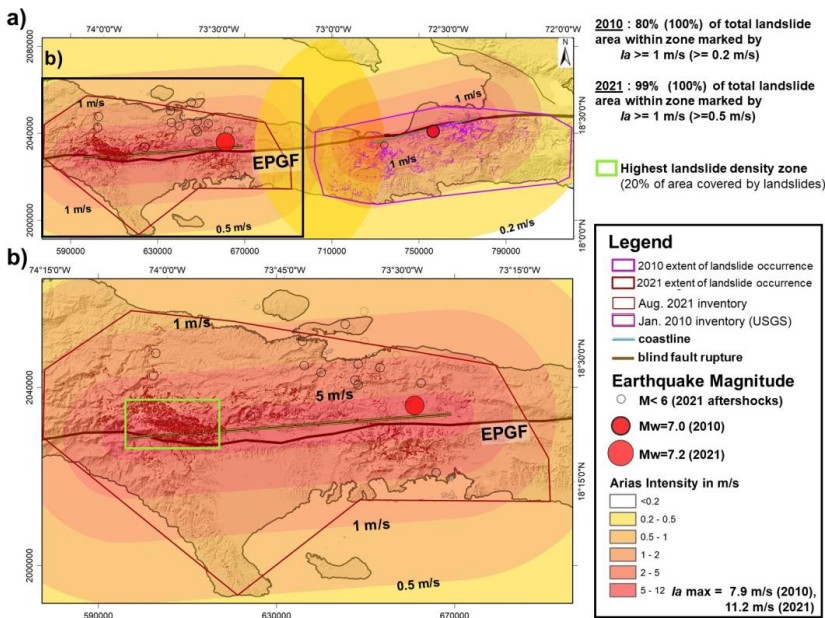


**Figure 8: a) Arias Intensity (Ia) attenuation maps computed (by using Eq. 7b, by Keefer and Wilson, 1989)**

**for the 2010 and 2021 main shocks in Haiti; see also indication of % of total surface area of landslides observed**

**for different Ia thresholds. b) Focus on the respective map computed for the 2021 event.**


To answer this and other related questions, a full landslide susceptibility analysis has been completed
and will be presented in another paper. Here, only the possible links between landslide distribution the
aforementioned seismotectonic and climatic factors will be discussed.
**4 Discussion**
**4.1 Discussion about landslide distribution characteristics**
From the comparison of the two landslide catalogues (2010 and 2021), we could first infer that apparently
not a single landslide triggered in August 2021 occurred within the zone previously impacted by the 2010
event. There is a gap of about 10 km between the westernmost 2010 and the easternmost 2021 landslide
(see gap between the general outlines of the maximum extent of landslides triggered in 2010 and in 2021

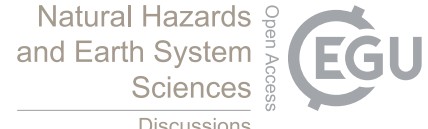

shown on the map in Fig. 4a). Thus, we assume that there was no obvious preconditioning of landslide
generation in 2021 by the 2010 event, while landslide studies completed in other parts of the World (e.g.,
by Parker et al., 2015, for events in New Zealand) could outline an influence of previous earthquakes on
landslide occurrence during later events. The absence of this influence by the 2010 earthquake is
probably due to the long distance (the 'gap') of about 60 km between the fault segments that ruptured in
2010 and in 2021. However, by citing Saint Fleur et al. (2020), Stein et al. (2021) hint at an older event,
of 1770, with an assumed magnitude of 7.5 and an epicenter located precisely in the gap between the
2010 and 2021 blind fault ruptures, which could also have affected the region hit by the 2021 earthquake.
At present, we cannot exclude that this older event had preconditioned some slopes (by soil weakening,
rock fracturing) affected by some larger landslides in 2021; however, very shallow slope failures initiated
in 1770 are unlikely to have stayed in place over such a long period of more than 250 years, as they
would have been 'washed' away by the next tropical rains.
Second, none of the two earthquakes triggered very massive landslides, such as deep-seated rockslides
with a volume of more than $10 \times 10^6$ m$^3$ (while extensive areas are covered by layered and weathered
limestone that could also produce massive slope failures; but this aspect will not be further discussed
here as the related geological influence on landslide occurrence will be analyzed in the landslide
susceptibility paper presently under preparation). Such massive failures have been observed after many
M7+ events in other mountainous regions of the world: see Fan et al. (2018) for the 2008 Mw=7.9
earthquake in China, or Havenith et al. (2015) for a series of M>7 events that hit Central Asian mountain
regions during the last 120 years. This is partly due to the fact that the regions hit by the two earthquakes
in Haiti are represented by mountains of limited elevation changes, typically less than 1000 m – while,
for instance, the Longmenshan Mountains hit by the 2008 Wenchuan earthquake, present elevation
changes of up to 3000 m over relatively short (<6 km) distances (Fan et al., 2018). This fact, combined
with the higher magnitude of the Wenchuan earthquake (Mw=7.9), could partly explain the much larger
number of massive rockslides triggered by the latter event in China. However, we have also to consider
some counterexamples of regions marked by mountainous relief that did not produce any very massive
rockslides during high-magnitude earthquakes (just like the 2010 and 2021 Haiti events), such as the part
of the Himalayas hit in 2015 by the Gorkha earthquake (see Lacroix, 2016). Thus, this problem related
to the more or less likely occurrence of massive rockslides in regions hit by high-magnitude earthquakes



is relatively complex, and cannot purely be approached by spatial analysis, as the one presented here;
more extensive numerical simulations would actually be necessary (but are definitely not the target of
our studies in Haiti) to assess the potential of seismically induced rockslides, such as those presented in
Gischig et al. (2015) or Lemaire et al. (2021).
Third, considering the values presented in Table 1, we still have to explain why the total surface area
covered by landslides in 2021 is much larger than the one covered by the 2010 landslides. We estimate
that this fact is likely to be related to (a combination of some of) the four following points: a) the first
likely physical reason for the larger area hit by mass movements in 2021 is the higher triggering
earthquake magnitude of the last event (this effect is also analyzed by comparing the influence of shaking
intensity on landslide distribution); b) another physical reason could be the possibly higher susceptibility
to mass movements of the western part of the peninsula hit by the 2021 event, compared to the eastern
part (this factor has to be analyzed on the basis of landslide susceptibility maps, considering also the
geological influence, which have been computed and will be presented in a follow-up paper); c) a third
reason for the larger area affected by landslides in 2021 could be related to the aforementioned 'hurricane'
effects that will be further discussed below; d) and fourth, the more central location of the fault segment
activated in 2021 with respect to the coasts of the peninsula certainly also explains parts of the larger
total surface areas of (subaerial) slope failures triggered during the last event within the wider onshore
hanging wall part, as already introduced above.
In this regard, we also highlighted the fact that the 2010 event triggered most landslides in the south of
the activated fault segment, while in August 2021 about 2/3 of all landslides were triggered in the north
of it. Considering the oblique slip character along the fault ruptures of 2010 and 2021 dipping to the
north, the hanging wall is located on the northside of the blind fault rupture projected on the surface -
(according to the fault mechanism provided by the USGS Earthquake Hazard Program page,
earthquake.usgs.gov). In this regard, the Wenchuan earthquake has clearly marked the effect of the
hanging wall on the landslide distribution: about 90 % of all landslides were triggered on top of the
reverse fault dipping towards the west-northwest, only a minor portion occurred on the more 'stable' foot
wall (Gorum et al., 2011; Fan et al., 2018). The 'hanging wall effect' on landslide triggering can be
explained by stronger upward oriented shaking that contributes to a higher surface acceleration and more
intense slope failures; additionally, all (or most of the) aftershocks occurred within the hanging wall,



increasing the seismic shaking intensity cumulated over the active seismic period in the related surface
area, which could have contributed to prolonged landslide activity as well (to be added to the climatic
effect introduced above and discussed below). This effect may thus also be at the origin of the more
widespread landslide occurrence in the north of the 2021 blind fault rupture. The reduced number of
'subaerial' landslides induced on the hanging wall side of the 2010 fault rupture can be explained by the
relative proximity of the respective fault rupture to the coast in the north and the absence of high and
steep slopes (onshore) on this side. Actually, a few known massive landslides occurred near the coast,
but are mostly located on submarine slopes in the 2010 hanging wall zone. Three of them reportedly also
caused tsunami waves (see Olson et al., 2011, among others) – a phenomenon that was not observed for
the 2021 event, as the coasts are located farther away from the seismic source zone.
**4.2 Discussion about landslide size-frequency characteristics**
Above, we clearly outlined the incompleteness of our 2021 inventory, for landslides smaller than about
3000 m$^2$; thus, it is likely that thousands of smaller landslides could not be mapped from the medium-
resolution Sentinel-2 imagery (10 m) and the higher resolution imagery (0.5 – 1 m) available on GEPro
for 50% of the target area before the end of 2021. To refine our landslide mapping in future, higher
resolution imagery must be used for the whole area affected by the 2021 event, and automatic landslide
identification techniques shall be applied by combining image analysis and machine learning as proposed
by Amatya et al. (2021). Actually, the 'manual' mapping applied now would take too much time to outline
the many thousands of very small landslides that have not been identified so far. Those would contribute
to the increase of the weight of the smaller landslides in the 2021 inventory, especially of those smaller
than 3000 m$^2$.
The incompleteness of the inventory notably limits its use for size-frequency analyses. However, above
we still presented related statistics and compared them with those made for the 2010 landslide inventory
to point out the clearly higher numbers of larger landslides triggered by the last event, compared with the
one in 2010 (based on 'landslide size' classes, for which the 2021 inventory can be considered as
complete). Actually, landslides triggered in 2010 mainly consisted of narrow slides and flows in
weathered limestone rocks, while the 2021 earthquake also induced landslide processes over wider slope
areas – as clearly shown by the 2021 and 2010 landslide zone views presented in Fig. 5; in the *Ravine*



*du Sud*, even entire slope units had failed in August 2021 (but the failed parts were typically not very
thick, less than 10 m).

**4.3 Discussion about climatic pre-conditioning effects**

We estimate that the different climatic conditions observed before the respective events may partly
explain the more widespread occurrence of larger landslides related to the 2021 event. In this regard we
indicated that the climatic contribution to landslide activity in 2021 might be twofold: first, some post-
seismic intensification of slope failures could be observed after the tropical storm Grace event that had
crossed the earthquake region on August 16-17, two days after the main shock. However, related effects
cannot really be quantified as only 10% of the total surface area potentially affected by the earthquake
appeared as cloud-free on imagery available right after the August 14 main shock and before August 16
(Grace event). For those limited areas, we estimate that storm Grace caused a widening of about 10-15%
of all slope failures with respect to the purely earthquake-induced landslide activation. Second, by
comparing the 2016-2017 landslide distribution with the one observed after August 14, 2021, it can be
seen that most of the October 2016 – end 2017 landslides occurred within the same region as the 2021
ones and most were clearly reactivated by the seismic shaking in August 2021 (while also many of them
had been revegetated in between). Above we could show that Hurricane Matthew had crossed the western
part of the peninsula in October 2016, producing an abnormal amount of precipitation precisely over the
area that was later hit by the earthquake (see GPM maps in Fig. 7), and where also hundreds of landslides
had occurred just after mid-October 2016. Therefore, it is very likely that this climatic event has triggered
many (and probably most) of the 625 mapped pre-seismic (October 2016 - pre-2018) landslides, which
preconditioned slope instability all over the area hit by the 2021 earthquake. Preconditioning of the
August 2021 slope failures could have been related to rock weakening and fracturing, and removal of the
protective vegetation cover during the 2016 Matthew event. Indeed, practically all 625 mapped October
2016 – pre-2018 landslide zones (at least 90% of them – and, considering that only for 50% of the entire
potentially affected area in 2016 landslide could be mapped over cloud-free zones) are located within the
landslide areas mapped for the August 2021 seismic event (which are still marked by a much higher level
of denudation compared to the October 2016 activation). The double hurricane effect (by Matthew in
2016 and by Grace just after the 2021 main shock) observed in the area hit by 2021 earthquake could be



responsible for the proportionally larger size of the 2021 landslides (estimating that the 2016 event, due
to its extreme intensity, made the strongest contribution) compared with the 2010 ones. In addition, we
have to consider that the 2010 earthquake had not been preceded by any particular hurricane event during
the previous ten years, at least not by any storm that had caused abnormal precipitation amounts (similar
to those caused by Hurricane Matthew) within the region hit by the 2010 earthquake.
Furthermore, the combined seismic and climatic influence could also explain the very different spatial
landslide distribution characteristics of the 2010 and 2021 catalogues: the relative dispersion of
landslides observed after the 2010 event could thus be partly related to the spatially highly variable effect
of tropical storms and hurricanes on landslide activity (acting over a longer period, with an influence that
could last over tens of years), partly overprinting the more concentrated seismic effect (resulting in
clusters of mass movements near the seismic source zone). The same dispersion might also have been
observed for the 2021 event if the central part of the seismically affected area had not been hit by that
major climatic event just five years before – doubling the landslide concentration effect in that area
(specifically for the 2021 event). However, we acknowledge that a quantification of these opposite effects
of climatic events, both on landslide dispersion and on their concentration, requires a more detailed
analysis, also focusing on specific sites, by completing numerical simulations of mass movements
affected by variable climatic (modelling changing groundwater level) and seismic influences (including
the effect of rock structures and types of lithologies and morphologies on shaking polarization and
amplification). A related landslide spatial distribution analysis should then also consider the influence of
extensive deforestation on slope destabilization, all over the country of Haiti. Actually, deforestation is
responsible for the decrease of 90% of the primary forest over the last few tens of years, especially in the
southern regions of Haiti where the two earthquake events had occurred (see Hedges et al., 2018). As
mostly shallow landslides occurred in 2010 and 2021, the effect of deforestation on the destabilization
of shallow soils and weathered rock cover must be taken into consideration for landslide occurrence
prediction. Such an extensive study would thus require the creation of an integrated seismotectonic-
morpho-geological-climatic-soil cover model allowing us to fully understand changing landslide activity
in Haiti – which is not the target of the present analysis (but will be partly approached in the follow-up
paper).





**4.4 Discussion about the regional seismic shaking influence on landslide distribution**

As for the climatic part, here, we only present regional data to outline some general seismic influences on landslide activity induced by the 2010 and 2021 earthquakes. Related maps (Fig. 8) show that the aforementioned gap of landslides between the areas affected by the earthquakes in 2010 and 2021 would indeed be marked both by a lower shaking intensity (but showing values that are still larger than the threshold Ia values observed elsewhere for landslide occurrence) and lower landslide susceptibility (a result that still has to be published). In the annex (Fig. A1), we also present the shakemaps produced by the USGS for the two events, but we did not compare landslide distributions with these maps as the latter do not seem to be coherent with respect to each other, noting that much larger intensities would have been produced by the lower magnitude event of 2010. Actually, it should be considered that such maps are also influenced by regional site effects (mostly on flat areas) that are not really relevant for landslide trigger mechanisms, and are also depending on ground measurements of seismic intensity that had not been well constrained during the 2010 due to missing seismic stations in Haiti at that time (a problem that starts to be solved now).

**5    Conclusions**

In this paper we presented a new landslide inventory created for the Mw=7.2 Nippes earthquake that occurred on August 14, 2021, in Haiti. Related spatial and statistical characteristics have been compared with those of the landslides mapped by others for the previous, Mw=7.0, January 12 (2010), earthquake that had occurred along the same fault zone (EPGF zone) but more to the East. Considering a series of uncertainties affecting the landslide statistics (related to the mapping technique, including the uncertain number of particularly small landslides triggered in 2021) and the environmental information (including some general climatic and geological conditions), this comparison allowed us to highlight the following points: 1) the 2021 earthquake triggered clearly bigger landslides than the one in 2010, and also the sum of all landslide areas is much larger than the one computed for the 2010 event; 2) a climatic preconditioning of slope instability could be outlined for the 2021 event, mainly in connection with the impacts of recent hurricanes in the 2021 affected region, which could also partly explain the more extensive landslide activity observed in 2021; 3) the 2010 landslides seem to be more dispersed around





the epicentral area than the 2021 slope failures, which could be due to the opposite climatic effect
inducing spatially more variable slope destabilization (also as no particular storm had hit the 2010
affected region just before or after the seismic event, as it was the case in 2021); this dispersion effect
can also be enhanced by the spatially varying deforestation that is locally very intense in the target areas.
We estimate that this proof of a likely combined seismic and climatic influence on landslide activity
(possibly augmented by morpho-geological and soil cover effects not studied in detail here) opens new
avenues for geohazard research, especially for regions like Haiti that are regularly hit both by severe
earthquakes and weather events. We also think that preconditioning of slope failures by multiple events
over longer terms, including by former earthquakes, should be studied more in detail as this
preconditioning could highly contribute to local and regional landslide hazards, both over short and
longer terms. A full analysis of such a scenario would require the development of an integrated
seismotectonic-morpho-geological-climatic-soil (and vegetation) cover model, combing extensive
spatial analyses with detailed numerical simulations, which can only be completed through an extensive
international multi-disciplinary collaboration around this target – which is obviously missing for Haiti.
Assessment of related risk would further require the involvement of experts in social geography and
economy. Also, a closer collaboration between scientists and the population shall be promoted as
recommended by Calais et al. (2022) and von Hillebrandt-Andrade and Vanacore (2022). Only when
these goals are achieved, we could really work on the prevention of at least parts of another future
earthquake disaster in Haiti.

**Acknowledgments**
This study was partly supported by the 'Earthquake Hazard and Vulnerability assessment – developing
innovative solutions for sustainable Risk Reduction and Communication in Haiti' project funding (2019-
2024) provided by the Belgian ARES – ACADÉMIE DE RECHERCHE ET D'ENSEIGNEMENT
SUPÉRIEUR.




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

**Annex**

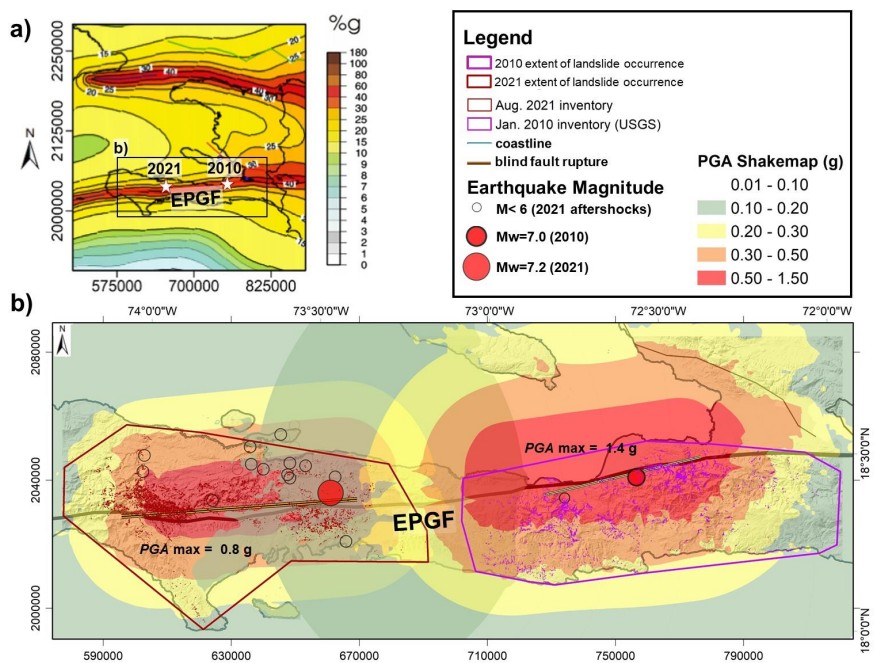

**Figure A1: a) Seismic hazard map of Haiti (modified from Frankel et al., 2011) with location of the January**
**12, 2010, and August 14, 2021, main epicenters. b) Combined overlays of shakemaps of the 2010 (right part**
**of map) and 2021 (left part) earthquakes.**





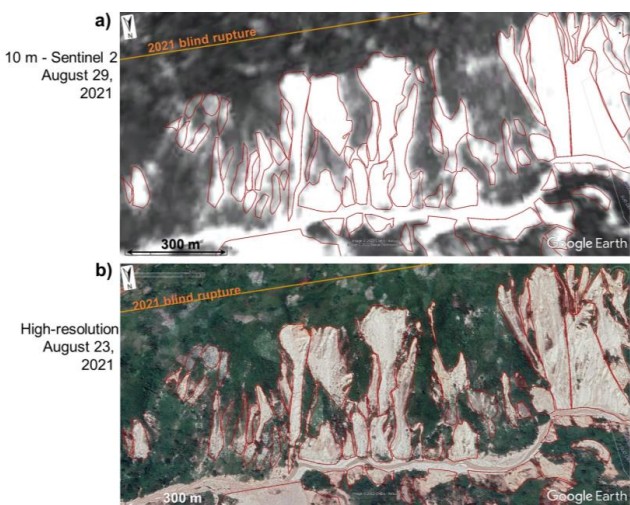


**Figure A2: Comparison between (a) a Sentinel-2 image (10-m resolution) and (b) a high-resolution (~0.5-1 m)**

**image (© Google Earth) of the same area affected by landslides triggered by the earthquake event in August**

**2021.**

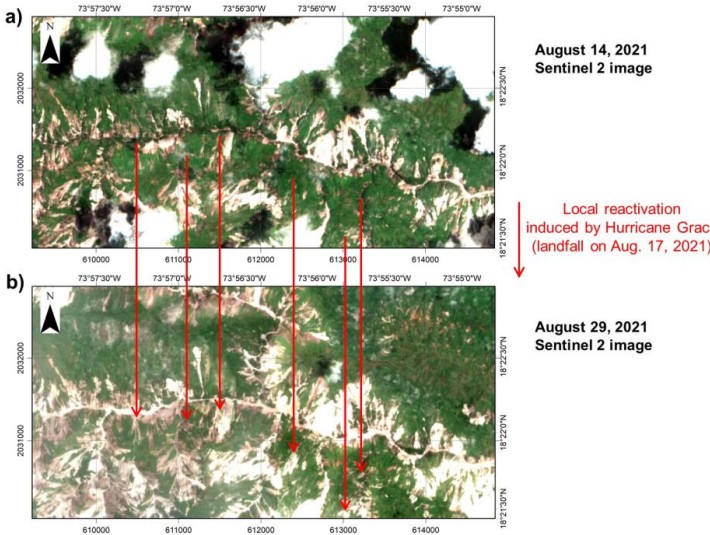


**Figure A3: Comparison between Sentinel-2 images (10-m resolution) for the same area obtained for (a) August**

**14 (about 2h after the main shock) and for (b) August 28, 2021 (12 days after impact by Hurricane Grace that**

**crossed the region on August 16, 2021). Red arrows point to zones where an intensification of denudation and**

**sliding can be observed.**