# Peer review of "Earthquake-induced landslides in Haiti: analysis of seismotectonic and possible climatic influences"

_Natural Hazards and Earth System Sciences, 2022_

## Referee Comment (RC2)

[referee-annotated manuscript omitted]

---

## Author Response (AR1)

Answers to reviewer requests/comments

Dear editor,

We would like to submit the revised version (with indicated track changes and without) of our manuscript: **Earthquake-induced landslides in Haiti: analysis of seismotectonic and possible climatic influences (by Havenith et al., subm.).**

Sincerely yours

Hans-Balder Havenith

Please, find below the answers to the different requests made by the two reviewers of our manuscript.

Note that we also had to include new information in the paper, as another paper (by Zhao et al., 2022 – see adapted references) has just published (in Geomorphology) similar data on the 2021 Haiti landslides. Comparison of statistical parameters is now included (actually just extending the existing analysis).

Also, note that in the track-changes version the whole first part looks like rewritten – actually, we have just rearranged the initial unique introduction section into two introduction & data section (moving paragraphs ..); the only new element includes 5 lines about he EPGF tectonics as requested by reviewer 1.

Reviewer 1 comments :

1) The epicentres in both the earthquake events; Aug. 14, 2021 earthquake (Mw 7.2) and Jan. 12, 2010 earthquake (Mw 7.2) are related to Enriquillo-Plantain-Garden Fault (EPGF), which is crucial in the genesis of these earthquakes. Though authors have described the EPGF briefly in line no. 60-70, page no. 3, a thorough tectonic description of the EPGF can be presented as supplementary annexure.

HBH: thanks, for the comment – yes, 5 lines about EPGF tectonics and earthquake history have been added in the new data section (section 2).

2) In Fig. 7, authors have presented climatic, particularly rainfall spatial variability. Fig. 7bc showing October rainfall variability follow the hypothesis relating Oct. 2016 hurricane with the landslides generated in Oct. 2016 and then reactivated at larger scale in 14, 2021 earthquake (Mw 7.2). However, it would be better if the authors present rainfall spatial variability of each month to justify the relative dominance of Oct. month particularly in 2016.

HBH: well, presenting these monthly data would certainly be interesting, but are not feasible in the present context (actually, average monthly data are presented in Fig. 7!). However, we included new precipitation data for August 2021 – a month marked by the activity of another hurricane, Grace – and which clearly shows that the earthquake-hit region had been affected by much more rainfall in October 2016, in connection with Hurricane Matthew, than with Hurricane Grace.

3) Authors have mentioned the landslide volume in absolute numbers in several statements throughout the MS. Please explain how you determined the volume of these landslides?

HBH : right, we used the equations we had introduced previously in a former paper. Therefore, we added a note in section 3.1.1 under Fig. 2 : ' – *note, for volume calculations, we used the equations proposed by Havenith et al., 2015, using as input the landslide deposit thickness estimate and surface area measurements'.*
The related reference was already included in the list.

Reviewer 2 comments :

1) General comments : The paper is generally well structured, but there are sometimes paragraphs which should better fit other chapters (especially related to the methodology and discussion-based ones, as outlined in the annotated .pdf).

HBH: see detailed answers to comments that were included in the annotated manuscript

As well, sometimes the paper has paragraphs which are difficult to be followed, due to the length of the sentences and the permanent comparisons between events. Sometimes, the authors are using sentences looking like a report (e.g. pages 6-7).

HBH: right! So, we revised and split all paragraphs covering more than 3 lines.

The graphics part is conclusive, but the choice of close colours in landslide delineation might not be the best one.

HBH: A color revision could not be made as we had already tested all combinations previously and this one was clearly the best one – however, we increased the contrasts for all map-figures.

Also, field pictures (besides the RS ones) with debris flows (as recognized by the authors as being the most numerous landslides triggered by the 2021 earthquake; instead, only rock falls are shown) would enhance the reader's overall image.

HBH: the reviewer is right- it would have been interesting to present some images of the most widespread types of landslides, the debris slides. However, all those regions were not accessible (also due to the hurricane event), and along the roads 'only' rock falls had occurred. That's why we show these ones – and also include a more detailed view of debris slides (by Google Earth Pro®) in Fig. A2.

2) Detailed comments included in the annotated manuscript (here only those beyond simple corrections that had all been respected):
   a) Numerous paragraphs seem to be more suitable to data-methodology section

   HBH : The introduction section has been split into 'introduction' presenting the general information and small literature review and into 'Regional ..' (a new section) to answer this request.

   b) Streets reopening was this the reason

   HBH: the 'reason' part has been removed as not necessary in this context.

   c) A rough size estimate would be helpful

   HBH: we include now : ' landslides (especially those smaller than 2000 m$^2$, corresponding roughly to 4 by 5 pixels on a Sentinel-2 image)'

   d) Disregarding the effect of the aftershocks .. 'is this the reason for this approach' ?

   HBH : we completed the information by : 'disregarding here the possible additional influence of aftershocks occurring at the same time in the region that could not be checked due to missing new high-resolution imagery after each event; it should also be noted that none of the aftershocks had a magnitude, Mw, larger than 6).'

   e) What about the time of 1$^{st}$ time failure/reactivations of former landslides?

   HBH: we specify now : 'However, we could at least see that no new landslides had been triggered within the earthquake-affected cloud-free zones. Effects of Hurricane Grace outside the region marked by stronger shaking have not been studied – this would require a completely new mapping approach.'

---

## Author Response (AR2)

Answers to reviewer requests/comments

Dear editor,

We would like to thank you for the proposed corrections and re-submit the newly revised version (with indicated track changes and without) of our manuscript: **Earthquake-induced landslides in Haiti: analysis of seismotectonic and possible climatic influences (by Havenith et al., subm.).**

Sincerely yours

Hans-Balder Havenith

Please, find below the answers to the different requests made by the editor's comments of our manuscript:

- line 168 : add a parenthesis after "measurement";

HBH: corrected.

- line 269, section 3.2 : it appears "strange" to start this paragraph with "in sub-section 4.2". Why not first describing the purpose of this section 3.2 with a sentence and then use sentences such as the ones in lines 360-361 to stress that results will be presented in sections 4.2 and 4.4 ?

HBH: right – was strange , we corrected it, but not exactly as you propose (we need this reference for introducing the 'equation' part. 'Section 4.2' was moved to the second sentence.

- lines 377 to 381 / line 576 : please check the font of the writing (it looks smaller);

HBH: right – corrected.

- line 936 : "combining" instead of "combing".

HBH : right – corrected.